# Overall Survival Rate in Allogeneic Stem Cell Transplanted Patients Requiring Intensive Care Can Be Predicted by the Prognostic Index for Critically Ill Allogeneic Transplantation Patients (PICAT) and the Sequential Organ Failure Assessment (SOFA) Scores

**DOI:** 10.3390/cancers14174266

**Published:** 2022-08-31

**Authors:** Adrien De Voeght, Evelyne Willems, Sophie Servais, Laurence Seidel, Michelle Pirotte, Paul Massion, Nathalie Layios, Maguy Pereira, Benoit Misset, Jean-Luc Canivet, Yves Beguin, Frédéric Baron

**Affiliations:** 1Department of Medicine, Division of Hematology, Centre Hospitalier Universitaire de Liège, University of Liège, 4000 Liège, Belgium; 2Department of Biostatistics, SIMÉ, University Hospital Center of Liège, 4000 Liège, Belgium; 3Department of Intensive Care, CHU and University of Liège, 4000 Liège, Belgium

**Keywords:** intensive care, prognostic score, allogeneic stem cell transplantation, PICAT, SOFA

## Abstract

**Simple Summary:**

Intensive care admission is a common complication of allogeneic hematopoietic stem cell transplantation. Mortality remains very high, and among several prognosis tools, data about power of discrimination showed contradictory results. The aim of our retrospective study was to evaluate the efficacy of a new score, the Prognostic Index for Critically Ill Allogeneic Transplantation (PICAT) Patients, for this specific setting in a cohort of 111 patients. We confirmed the ability of this score to discriminate three groups of patients with different outcomes. Moreover, we evaluated a classic intensive score, the Sequential Organ Failure Assessment (SOFA), and we showed that the SOFA outperformed the PICAT to predict outcomes in our cohort.

**Abstract:**

**Background.** Allogeneic hematopoietic stem cell transplantation (allo-HCT) recipients requiring intensive care unit (ICU) have high mortality rates. **Methods.** In the current study, we retrospectively assessed whether the Prognostic Index for Critically Ill Allogeneic Transplantation patients (PICAT) score predicted overall survival in a cohort of 111 consecutive allo-HCT recipients requiring ICU. **Results.** Survival rates at 30 days and 1 year after ICU admission were 57.7% and 31.5%, respectively, and were significantly associated with PICAT scores (*p* = 0.036). Specifically, survival at 30-day for low, intermediate, and high PICAT scores was 64.1%, 58.1%, and 31.3%, respectively. At one-year, the figures were 37.5%, 29%, and 12.5%, respectively. In multivariate analyses, high PICAT score (HR = 2.23, *p* = 0.008) and relapse prior to ICU admission (HR = 2.98, *p* = 0.0001) predicted higher mortality. We next compared the ability of the PICAT and the Sequential Organ Failure Assessment (SOFA) scores to predict mortality in our patients using c-statistics. C statistics for the PICAT and the SOFA scores were 0.5687 and 0.6777, respectively. **Conclusions.** This study shows that while the PICAT score is associated with early and late mortality in allo-HCT recipients requiring ICU, it is outperformed by the SOFA score to predict their risk of mortality.

## 1. Introduction

Despite its high morbidity and mortality rate [1], allogeneic hematopoietic stem cell transplantation (allo-HCT) has remained a potentially curative option for many patients suffering from malignant or non-malignant hematologic disorders [2] and is still the best curative option for acute leukemia [3,4]. While 10 to 50% of transplanted patients require intensive care (ICU) transfer as treatment of life-threatening complications, reported intensive care survival rate for allo-HCT recipients admitted to ICU has remained poor, being less than 50% at day 30 and 20% at one year [5,6,7,8,9]. It is thus particularly important to identify categories of allo-HCT recipients who have a better chance for survival once admitted to the ICU.

Several scores have been shown to predict mortality of allo-HCT recipients admitted to the ICU. Gilli et al. reported that the Sepsis-Related Organ Failure Assessment (SOFA) score, one of the most commonly used ICU scores, predicted mortality in a cohort of allo-HCT patients admitted to ICU [10]. Another study found that the pre-transplant hematopoietic cell transplantation comorbidity index (HCT-CI) [11] predicted survival of allo-HCT patients requiring ICU [7]. More recently, the PICAT (Prognostic Index for Critically Ill Allogeneic Transplantation Patients) index has been developed as a specific index for critically ill allo-HCT patients [12]. This score was developed by Bayraktar et al. and is based on different parameters such as clinical features, biological data, and timing and reason for ICU admission. One of the main advantages of this score is that it can be calculated before ICU admission.

The ongoing coronavirus disease 2019 (COVID-19) pandemic and its dramatic need for ICU facilities has been a real challenge for all ICU and hematology teams [13]. Therefore, when resources are short, admission of non-COVID patients with a poor to very poor 1-year survival can be debated. Facing these difficult decisions, the use of the most adequate score may help medical teams to decide which patients would optimally benefit from an available ICU bed.

In the current study, we assessed the 30-day and 1-year survival rates of allo-HCT patients requiring ICU transfer at our center and assessed the ability of the PICAT index to predict it. We next compared the ability of the PICAT and the SOFA scores to predict mortality of allo-HCT patients admitted to the ICU.

## 2. Materials and Methods

### 2.1. Methods and Definitions

This mono-centric study is a retrospective analysis of allo-HCT patients admitted between January 2008 and December 2018 to the intensive care units of a tertiary hospital (Centre Hospitalier Universitaire of Liège, Belgium).

Demographic data such as age, sex, pre-transplant comorbidity, Karnofsky performance status, disease and transplantation data, ICU data, and follow-up were extracted from medical charts.

Transplant-related parameters included: underlying disease, donor type, time from allo-HCT to ICU admission, conditioning intensity (reduced-intensity (RIC) versus myeloablative (MAC) regimens classified according to EBMT guidelines [2], and HCT-CI score [11]. Parameters collected at the time of ICU admission included presence or absence of an infection and/or graft-versus-host disease (GVHD), ongoing systemic immunosuppression, platelet transfusion dependency, cause of ICU admission, and biological data (complete blood count, renal function, liver function, coagulation parameters, and colonization or not with a multi-drug resistant bacteria). Relapse at the time of ICU admission was defined as the relapse of the original disease or disease progression according to the definition from EBMT guideline [2].

Grading of GVHD was made according to previously published international guidelines (acute GVHD [14] and chronic GVHD [15]) and made by a senior member of the transplantation team.

Neutropenia referred to a neutrophil count below 0.5 × 10^9^/L. Sepsis was diagnosed according to the most recent guidelines [16]. Acute respiratory failure was defined as failure of oxygen support with nasal cannula to maintain acceptable pulse oximetry.

Disease risk index (DRI) [17], SOFA [18], HCT-CI [11], and PICAT [12] scores were calculated as previously described. Specifically, PICAT score is a computing score with several items with different coefficients. Parameters included in the PICAT score consist of time from hospital admission to ICU (1.56 if >30 days), LDH ≥ 2 × ULN (1.53), bilirubin ≥ 2 mg/dL (1.24), albumin < 3 g/dL (1.24), acute respiratory failure as cause of ICU admission (0.97), prothrombin time-international normalized ratio >2 (0.91), MAC regimen (0.67), age above 60 years (0.43), and HCT-CI comorbidity score ≥2 at transplantation (0.27). Parameters calculated at the time of ICU admission included biological parameters (LDH, albumin, bilirubin levels, and prothrombin time-international normalized ratio), and age.Based on their PICAT index, patients were distributed into 3 categories as previously described [12]: low (0–2), intermediate (2–4), and high (≥4) scores. Based on their SOFA scores, patients were distributed into 6 categories as previously described [18]. For patients with multiple ICU admissions, PICAT index and subcategorization were determined at the time of the first admission only.

Cause of death was determined following previously described criteria [19].

### 2.2. Statistical Methods

Qualitative variables were reported as frequencies and continuous variables as median and range.

Overall survival (OS) was defined as survival from the day of ICU admission to the day of death or the date of last contact. OS probabilities were calculated using the Kaplan–Meier method. For patients with multiple stays at ICU, only the first stay was considered, and patients were not censored at the time of the next ICU admission. To compute OS with several parameters (uni- and multivariate), Cox regression models were used. A step-wise backward procedure (included any covariates with a *p*-value of <0.15 on univariate analysis) was used to build a set of independent factors for 30-day and 1-year mortality after ICU admission in addition to the PICAT score (forced in the Cox models). C-statistics were computed for assessment of model performance. Results with a *p*-value under 0.05 were considered significant, and all *p*-values were two-sided. Statistical analyses were performed with SAS version 9.4 and figures with R version 3.6.1.

## 3. Results

### 3.1. Patients

The records of 535 consecutive allo-HCT patients performed in 502 patients between January 2008 and December 2018 at our center were reviewed. A total of 111 patients required ICU (Table 1). Median follow-up for all patients after first ICU admission was 50 months (range from 0 to 136 months), and median follow up for surviving patients was 87 months (range from 22 to 136 months).

Median age at admission was 56 years (range, 7–71 years), with 66.4% (*n =* 74) of the patient being younger than 60 years (Table 1). Acute leukemia and myelodysplastic syndrome (MDS) represented the major indications for allo-HCT (59.4%). Thirty-two (28.8%) patients had a Karnofsky status below 80 at transplantation. Donors included human leukocyte antigen (HLA)-identical siblings (*n =* 24, 21.6%), one HLA-mismatched related donor (*n =* 1, 0.9%), 10/10 HLA-matched unrelated donors (*n =* 49, 44.2%), mismatched unrelated donors (*n =* 22, 19.8%), HLA-haploidentical donors (*n =* 9, 8.1%), and unrelated cord blood (*n =* 6, 5.4%). The HCT-CI score was <2 in 42.3% of the population, and median HCT-CI score at transplantation was 2 (range from 1 to 8). Before ICU admission, 17 patients (15.3%) of the cohort had a relapse of their underlying disease. Sixty-one (55%) needed platelet transfusion, and fourteen (12.6%) carried multi-drug resistant bacteria (MDR-bacteria). Finally, 21 patients (18.9%) had acute graft-versus-host disease(GVHD) (90.5% with grade ≥2) on admission, and 9 (8.1%) had ongoing chronic GVHD (3 patients had mild, 5 had moderate, and 1 had severe chronic GVHD according to National Institutes of Health (NIH) classification [15]).

Sixty-four patients (57.7%) had low PICAT scores; thirty-one (27.9%) had intermediate scores; and sixteen (14.4%) were high risk. SOFA score was subdivided into 6 categories of severity (*n =* 109 patients): SOFA 0–6 (*n =* 45, 41.3%), SOFA 7–9 (*n =* 33, 30.3%), SOFA 10–12 (*n =* 15, 13.8%), SOFA 13–14 (*n =* 9, 8.2%), SOFA 15 (*n =* 3, 2.7%), and SOFA 16–24 (*n =* 4, 3.7%). Among positive items of the PICAT score, respiratory failure as the reason for ICU admission was the preponderant positive item for 82 patients (73.9%). This was followed up by high HCT-CI score (for 64 patients, 57.7%), low albumin (41 patients, 36.9%), age and MAC (both positive for 37 patients, 33.3%), then elevated LDH (26 patients, *n =* 23.4%), elevated bilirubin (23 patients, 20.7%), longer time between hospital admission and ICU admission (17 patients, 15.3%), and finally elevated PT-INR (3 patients, 2.7%).

### 3.2. ICU Admission

A total of 111 patients (22.1% from our original cohort of 502 allo-HCT patients) was admitted to the ICU and required one (*n =* 82), two (*n =* 21), three (*n =* 5), or four (*n =* 2) ICU admissions, respectively. The distribution of the rate of admission for the first period (2008–2009) was 20.7%, for the second period (2010–2012) 27.9%, for the third period (2013–2015) 26.2%, and for the fourth period (2016–2018) 25.2% (Table 2). At the time of the first admission to ICU, 98 patients (88.3%) had received one allo-HCT, 12 patients (10.8%) had received a second allo-HCT, and 1 patient (0.9%) had received a third allo-HCT.

Acute respiratory failure was the leading cause for ICU admission (44.2%) (Table 2). This was followed by sepsis (17.1%), neurologic failure (12.6%), digestive failure (7.2%), GVHD (4.5%), renal failure (3.6%), uncontrolled hemolytic anemia (1.8%), and other (9%—post cardiac arrest, postoperative care, and cardiogenic shock). The vast majority of the population was admitted before 180 days after allo-HCT (*n =* 92, 82.9%), and the median time between transplantation and admission was 56 days (range from −3 to 1050 days). Median PICAT at ICU admission was 1.74 (range from 0 to 6) with 64 (57.7%) patients having a low PICAT score (from 0 to 2), 31 (27.9%) an intermediate PICAT score (>2 to 4), and 16 (14.4%) a high PICAT score (≥4). Median SOFA score at admission was 7 (range from 2 to 18). Median SOFA scores at admission for patients with low (*n =* 64), intermediate (*n =* 31), and high PICAT (*n =* 16) score were 7 (range 2 to 18), 8 (range 2 to 15), and 10 (range 5 to 16), respectively.

### 3.3. OS

OS at 30 days and at 1 year after ICU admission was 57.7% and 31.5%, respectively (Figure 1A). For patients (*n =* 47) that died within the first 30 days after ICU admission, primary causes of death included infections (*n =* 14, 29.7%), relapse (*n =* 12, 25.5%), aGVHD (*n =* 9, 19.2%), multiple organ failure (MOF) (*n =* 4, 8.4%), cGVHD (*n =* 2, 4.3%), cerebral nervous system (CNS) failure (*n =* 2, 4.3%), hepatic failure (*n =* 2, 4.3%), and other (*n =* 2, 4.3%). For those that died from 30 days to 1 year after ICU admission (*n =* 29), primary causes of death included infections (*n =* 8, 27.6%), aGVHD (*n =* 8, 27.6%), relapse (*n =* 7, 24.1%), cGVHD (*n =* 2, 6.9%), CNS failure (*n =* 2, 6.9%), and hepatic failure (*n =* 2, 6.9%). We observed that the PICAT score predicted transplantation outcomes in univariate analysis (log-Rank test, *p* = 0.029; Figure 1B). Specifically, survival at 30 days for low, intermediate, and high PICAT score was 64.1%, 58.1%, and 31.3%, respectively. Survival at 1year for low, intermediate, and high PICAT was 37.5%, 29%, and 12.5%, respectively (Figure 1B).

In univariate Cox analyses, PICAT score (*p* = 0.036), platelet transfusion requirement before ICU admission (*p* = 0.0065), HLA—haplo-identical donor (*p* = 0.033), and disease relapse before ICU admission (*p* = 0.0001) predicted mortality. There was a similar trend for carrying MDR bacteria at ICU admission (*p* = 0.075) (Table 3). In contrast, the time period of ICU admission did not significantly correlate with mortality in our cohort (*p* = 0.13).

In multivariate analyses, high PICAT score (HR = 2.23, *p* = 0.008) and disease relapse before ICU admission (HR = 2.98, *p* = 0.0001) were independently associated with higher mortality (Table 4).

Since a part of the parameters of the PICAT score are related to the transplant itself, we evaluated if the peri-transplant period (before day 180, *n =* 92, 82.9%) was more affected than the later period (>180 days after transplantation, *n =* 19, 17.1%) with the PICAT score. We performed a proportional odds logistic regression between the three different categories of the PICAT and the period (peri-transplant vs. late). We did not find a difference in the PICAT score regarding the period (*p* = 0.41, OR = 1.46 [0.60–3.57]).

### 3.4. Prediction of Mortality following ICU Administration with the PICAT and the SOFA Scores

We next compared the ability of the PICAT and the SOFA scores to predict mortality of allo-HCT patients admitted to the ICU. First, we assessed whether there was a correlation between the PICAT and the SOFA scores and observed that the two scores were significantly correlated (r = 0.32; *p* = 0.0008; Figure 2).

We next compared the ability of the PICAT and SOFA scores to predict mortality in allo-HCT patients requiring ICU using c-statistics. We observed that c statistics for the PICAT was 0.5687 versus 0.6777 and 0.6577 for the SOFA score when assessed as a continuous or a categorical variable, respectively (Table 5). Further, interestingly, combining the PICAT and the SOFA scores did not significantly improve the performance of the SOFA score (c statistics = 0.6675).

## 4. Discussion

Overall survival of allo-HCT patients who need ICU admission remains very poor, down to only 20% at one year. Events associated with decreases of care resources could lead to questions being raised about fair allocations of precious ICU beds, especially for populations with very poor outcomes. Thus, the decision of ICU admission or not of allo-HCT patients should be based on objective parameters. Various scoring systems have been developed to predict mortality of patients admitted to the ICU: SOFA, Acute Physiology and Chronic Health Evaluation (APACHE), etc. [18]. Unfortunately, these scores cannot be calculated before ICU admission, somewhat limiting their interest [10,20,21]. Indeed, one item of the SOFA score is the oxygen fraction in the inspired air, which is difficult to evaluate at the bedside of the patient in non-ICU units. Moreover, the patient’s oxygen in arterial blood needs to be known. The APACHE score is computed from 12 physiologic variables measured within 24 h of admission and thus can only be defined after admission. Recently, PICAT, a specific ICU score for allo-HCT, has been developed by the MD Anderson group [12]. This score is based on clinical, biological, and allo-HCT data of the patient and is thus both easy to compute and computable before ICU admission, since it is not related to ICU procedures such as mechanical ventilation, hemodynamic support, and renal replacement therapy, which are well-known ICU predictors of mortality. This prompted us to perform a retrospective study assessing the ability of the PICAT score to discriminate outcomes of consecutive allo-HCT patients admitted to the ICU at our center. Moreover, we wanted to gauge performance of the SOFA score in our population. SOFA is a very easy score to compute and it does not require a stay of 24 h to be calculated. Several observations were made.

Our population is composed of patients who shared demographic data such as age (median age: 56 years) and severity (median SOFA score: 8) with other previously reported studies [8,20,22,23]. In a recent randomized controlled trial of ICU patients investigating new treatment strategies for septic shock and ARDS, the median SOFA scores were between 8.5 and 9.2 [24,25]. Thus, our patients shared the same severity as reference ICU populations. Nevertheless, in our population, RIC was preponderant (67.5%), which is consistent with a recent cohort [8,26]. Several other studies showed a higher proportion of MAC in their cohort [8,20,22,23]. These studies were performed earlier than ours, explaining this discrepancy. Indeed, the work of Lengline et al. [22] showed a modification of the proportion of patients requiring ICU care according to whether they were conditioned with MAC versus RIC. During the first period (1997–2003), 89% of the patients were conditioned with a MAC regiment, whereas only 54% received a MAC during the second period of analysis (2004–2007). The rate of ICU admission in our cohort (22.1%) tended to be higher than in other studies [20,24,27], where 10 to 15% of the population of transplant patients were admitted in the ICU. However, our population was in the range of admission described in a recent meta-analysis (10–53% admission rates between 1995 and 2012) [5]. As reviewed in that recent meta-analysis [5], OS has significantly improved over the last decade. Our survival rate of 31.5% at one-year post-ICU admission is consistent with these data. New strategies, some specific for managing acute respiratory failure in cancer patients [22,28,29] and others, relative to the management of hemodynamic instability [30] as the early-goal therapy, may explain these results. In addition, our study evaluated admission without focusing on the specific timing after transplantation, although it is well-described that the risk of ICU death is especially high in the early post-transplant period (i.e., before day 100) [5].

The first observation of our study was that the PICAT score could indeed discriminate three categories of patients with different survival. A high PICAT was associated with a 1-year survival rate of 12.5% versus 29% and 37.5% in patients with intermediate and low PICAT scores, respectively. This is in concordance with data from the original publication [12]. Our data thus confirm the interest of the PICAT score for predicting mortality in allo-HCT patients admitted to the ICU. These results, however, contrast with those observed in a recent analysis from the group of Mainz (Germany) in a cohort of 81 patients. In that report, the authors observed that the APACHE-II and SOFA scores, but not the PICAT scores, predicted ICU survival [27]. Specifically, ICU survival was 67%, 56%, and 54% in patients with low (*n =* 9), intermediate (*n =* 34), and high (*n =* 34) PICAT scores, respectively. Reasons for these discrepancies are unclear but might rely on the fact that the Mainz study was focused on young patients admitted to the ICU during the immediate post-transplant period and included a majority of patients conditioned with MAC (75%) and having very high mean SOFA of 14 and a PICAT score of 4 at ICU admission, which is not classical for this kind of specific population [5]. Moreover, we found a significant difference between our work and the study of Bayraktar, who developed the PICAT score [12]. In their study, relapse was not an independent predictor of death. In our study, relapse was an important mortality predictor.

A second important observation was that the SOFA score surpassed the PICAT score for predicting mortality in our patients. Although these two scores shared only two similarities: grading the level of bilirubin and respiratory failure (in the SOFA, with the rapport of oxygen fraction in the inspired air and the patient’s oxygen in arterial blood), we observed a correlation between the two scores. Although the original work of MD Anderson’s group found that PICAT score was significantly better than SOFA [12], our observation was consistent with that reported by the group of Mainz, who also observed that the SOFA was more powerful than the PICAT score to predict mortality of allo-HCT patients admitted to the ICU [27]. Interestingly, combining both scores did not significantly improve the performance of the SOFA score used alone, possibly because the two scores were correlated. Further, it should be noted that the predictive value of the combined score remained relatively low (c statistics = 0.6675), emphasizing that decision for ICU admission or not of allo-HCT patients should not be based on these scores alone.

Although there is no perfect score to evaluate risk of death or survival, these data highlight that patients with low scores, either PICAT or SOFA, should be admitted if necessary without discussion and without any limitation.

Recently, poor graft function (defined as cytopenia due to lack of production in at least two hematologic cell lineages) and low count of platelets have been demonstrated to be important predictors of mortality in allo-HCT patients admitted to the ICU [31]. In the original work of Bayraktar et al., thrombocytopenia (defined as platelets count < 50,000/µL) was associated with mortality in the univariate analysis [12]. Low platelet count was associated with poor outcomes in a population of allo-HCT patients outside ICU [32]. Univariate analyses in our cohort confirmed that low platelet count was associated with high ICU mortality. However, it was not found to be an independent factor in multivariate analysis, perhaps because of the relatively small sample size of our cohort.

As reported in a meta-analysis [5], the disease status at ICU admission is underreported and so poorly evaluated. In our multivariate analysis, relapse was an independent marker of death. Although this is in contrast to what has been observed in the original paper describing the PICAT score, our observation is not unexpected given the grim prognosis of patients relapsing after allo-HCT [33].

This study has several limitations. First, it is a monocentric retrospective analysis. Secondly, it includes only data of patients who were admitted to the ICU. We did not include patients who needed IC but refused or for whom admission was considered futile. In addition, this analysis includes data from before the COVID-19 crisis, and thus whether the PICAT score could predict outcomes of COVID-19 allo-HCT patients requiring ICU remains to be studied. Fortunately, anti-SARS-CoV-2 vaccination and the use of anti-SARS-CoV-2 monoclonal antibodies have substantially reduced the proportion of allo-HCT patients requiring ICU for COVID-19 in our center [34,35,36]. Thirdly, all admissions were included regardless of time after transplantation, which could partly explain the good results we report here compared with previously published papers.

## 5. Conclusions

This study confirms that the PICAT score can be easily used and might help to discriminate outcomes of allo-HCT patients who need ICU admission. It also confirms that the PICAT score is correlated with the SOFA score, and this last score outperformed the PICAT score to predict survival of this specific population.

## Figures and Tables

**Figure 1 cancers-14-04266-f001:**
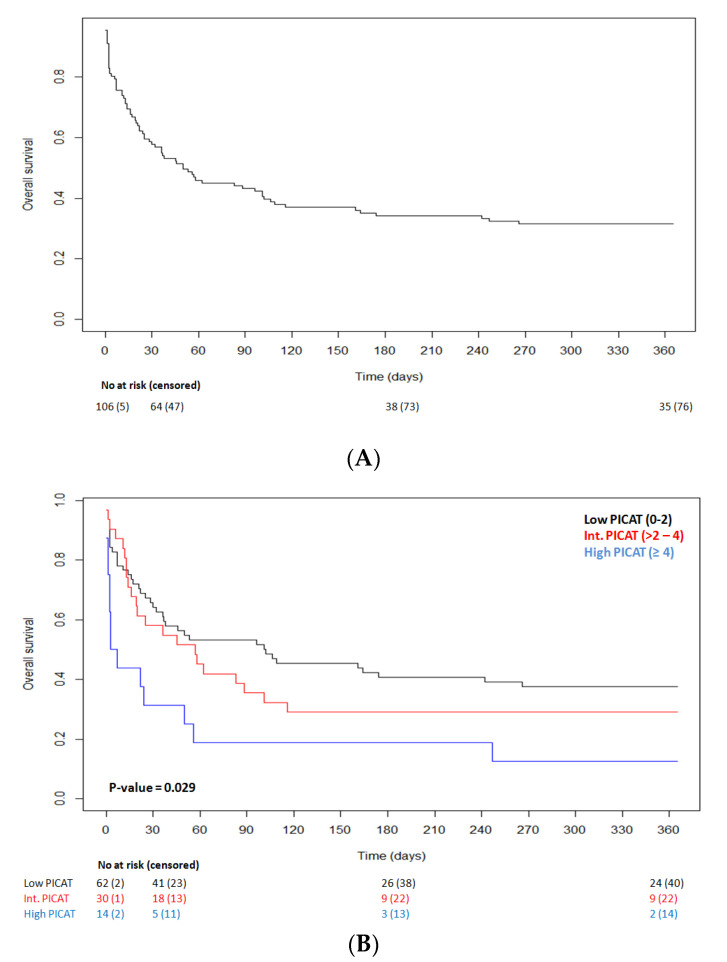
(**A**) Kaplan–Meier curve of overall survival after admission in ICU. Thirty-day OS is 57.7%, and 1-year OS is 31.5%. (**B**) Kaplan–Meier curve of OS according to PICAT classes (10); low in black (*n =* 64), intermediate in red (*n =* 31), and high in blue (*n =* 16); *p*-value = 0.029.

**Figure 2 cancers-14-04266-f002:**
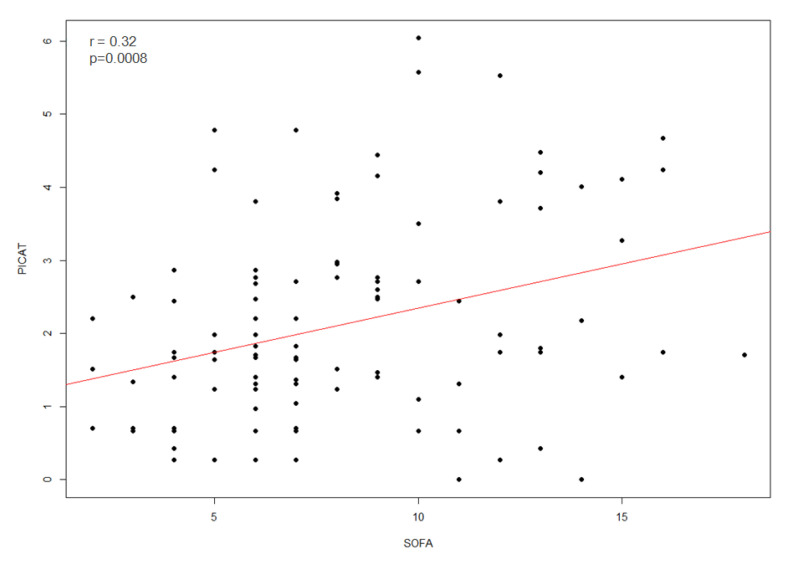
Correlation (red line) between PICAT score and SOFA score (*n =* 109) at time of admission in ICU with r = 0.32 and *p* = 0.0008.

**Table 1 cancers-14-04266-t001:** General characteristics of allo-HCT patients admitted to the ICU.

Characteristic	Number of Patients	
	*n =* 111	%
**Age**		
0–20	11	9.9
21–40	18	16.2
41–60	45	40.6
61–71	37	33.3
**Sex**		
Female	44	39.6
Male	67	60.4
**Disease**		
AML	36	32.4
ALL	16	14.4
MDS	14	12.6
NHL	10	9
Plasma cell disorder	6	5.4
CML	5	4.5
MPN	5	4.5
CMML	4	3.6
CLL	3	2.8
HL	2	1.8
Other	10	9
**Karnofsky at transplantation**		
90–100%	56	50.5
80%	23	20.7
<80%	32	28.8
**Conditioning regimens**		
RIC	75	67.6
MAC	36	32.4
**Donor HLA match**		
10/10	73	65.8
Mismatch	29	26.1
Haplo-identical	9	8.1
**Donor relationship**		
Unrelated	77	69.4
Related	34	30.6
**Donor Graft**		
MUD	49	44.2
M-SIB	24	21.6
MMUD	22	19.8
Haplo-id	9	8.1
CB	6	5.4
MM-SIB	1	0.9
**Platelet transfusion dependency at ICU admission**		
YES	61	55
NO	50	45
**MRD-bacteria carriers**		
YES	14	12.6
NO	97	87.4
**Acute GVHD**		
YES	21	18.9
NO	90	81.1
**Stage of acute GVHD**	*n =* 21	
1	2	9.5
2	6	28.6
3	8	38.1
4	5	23.8
**Chronic GVH**	*n =* 111	
Yes	9	8.1
No	102	91.9
**Relapse before admission**		
YES	17	15.3
NO	94	84.7
**Disease Risk Index** [17]		
Low risk	22	19.8
Intermediate risk	49	44.2
High risk	29	26.1
Very high risk	11	9.9
**HCT-CI** [11]		
0–1	47	42.3
≥2	64	57.7
**PICAT score** [12]		
0–2	64	57.7
>2–4	31	27.9
≥4	16	14.4
**SOFA score** [18]	*n =* 109	
0–6	45	41.3
7–9	33	30.3
10–12	15	13.8
13–14	9	8.2
15	3	2.7
16–24	4	3.7
**Variable of the PICAT score**	*n =* 111	
Time from hospital admission to ICU	17	15.3
LDH ≥ 2 × ULN	26	23.4
Bilirubin ≥2 mg/dL	23	20.7
Albumin < 30 g/L	41	36.9
Respiratory failure as the reason for ICU admission	82	73.9
PT-INR ≥ 2	3	2.7
MAC	37	33.3
Age > 60 years	37	33.3
HCT-CI ≥ 2	64	57.7

ALL = acute lymphoblastic leukemia, AML = acute myeloid leukemia, CB = cord blood, CLL = chronic lymphocytic leukemia, CML = chronic myeloid leukemia, CMML = chronic myelomonocytic leukemia, GVHD= graft-versus-host disease, Haplo-id = haplo-identical donor, HK = Hodgkin’s lymphoma, MAC = myeloablative conditioning, MDS = myelodysplastic syndrome, MPN = myeloproliferative neoplasms, MRD = multi-resistant drug, M-SIB = matched-sibling donor, MUD = matched-unrelated donor, MM-SIB = mismatched sibling, MMUD = mismatched unrelated donor, NHL = non-Hodgkin lymphoma, other = aplastic anemia, Fanconi anemia, monosomal genetic disorder, primary immunodeficiency, RIC = reduced-intensity conditioning.

**Table 2 cancers-14-04266-t002:** Causes, timing, and history of ICU admission.

Causes	Number of Patients	
	*n =* 111	%
Acute respiratory failure	49	44.2
Sepsis	19	17.1
Neurologic failure	14	12.6
Digestive failure	8	7.2
GVHD	5	4.5
Renal failure	4	3.6
Hemolytic anemia	2	1.8
Other	10	9.0
**Timing**	**Number of Patients**	
	*n =* 111	%
During conditioning	6	5.4
Between transplantation and engraftment	27	24.3
Between engraftment and day 30	12	10.8
Between day 30 and day 100	29	26.2
Between day 100 and day 180	18	16.2
After day 180	19	17.1
**History of admission**	**Number of Patients**	
	*n =* 111	%
2008–2009	23	20.7
2010–2012	31	27.9
2013–2015	29	26.2
2016–2018	28	25.2

**Table 3 cancers-14-04266-t003:** Univariate Cox-regression analysis of overall survival.

Variable	Categories	*N*	Hazard Ratio	95% Confident Interval	*p*-Value
Disease Risk Index		111			0.96
	Low risk		Ref.		
	Intermediate risk		1.068	(0.595–1.916)	
	High risk		1.045	(0.555–1.969)	
	Very high risk		1.248	(0.566–2.753	
HCT-CI score		111			
	0–1		Ref.		0.43
	≥2		1.188	(0.774–1.823)	
Karnofsky		111			
	90–100		Ref.		0.44
	80		1.203	0.698–2.073	
	<80		1.358	0.841–2.194	
RIC		111			0.18
	No		Ref.		
	Yes		1.376	0.865–2.188	
Age > 60 years		111			0.11
	No		Ref.		
	Yes		1.421	0.825–2.183	
PICAT classes		111			**0.036**
	0–2		Ref.		
	>2–4		1.257	0.779–2.028	
	≥4		**2.162**	**1.200–3.897**	
Platelet transfusion dependent		111			**0.0065**
	No		Ref.		
	Yes		**1.818**	**1.182–2.796**	
GVHD		111			0.61
	No		Ref.		
	Yes		**1.143**	**0.687–1.903**	
Donor		111			
	Related		Ref.		**0.033**
	Haplo		**3.238**	**1.339–7.829**	
	Unrelated		1.421	0.857–2.357	
HLA classes		111			0.46
	10/10		Ref.		
	Other		1.176	0.762–1.814	
MDR-bacteria carriage		111			0.075
	No		Ref.		
	Yes		1.716	0.948–3.107	
Relapse before admission		111			**0.0001**
	No		Ref.		
	Yes		**2.899**	**1.680–5.002**	

Bold is use for all significant data with a *p* value < 0.05.

**Table 4 cancers-14-04266-t004:** Multivariate COX model of prognostic factors of OS.

Parameter	Hazard Ratio	95% Confident Interval	*p*-Value
Low PICAT	Ref.		
Intermediate PICAT	1.132	(0.697–1.838)	0.62
High PICAT	2.23	(1.23–4.03)	**0.008**
Relapse	2.98	(1.71–5.19)	**0.0001**

Bold is use for all significant data with a *p* value < 0.05.

**Table 5 cancers-14-04266-t005:** Cox-regression and c-statistics analysis of mortality.

Variable	Categories	N	Hazard Ratio	95% Confident Interval	*p*-Value	c
**Univariate model**						
PICAT classes		111			**0.036**	0.5687
0–2			Ref			
>2–4			1.26	0.78–2.03		
≥4			2.16	0.78–2.03		
SOFA (continuous data)		109	1.14	1.08–1.2	**<0.0001**	0.6777
SOFA (categorical data)		109			**0.0005**	0.6577
1 (0–6)			Ref			
2 (7–9)			2.37	1.40–3.99		
3 (10–12)			2.52	1.30–4.88		
4 (13–14)			3.97	1.84–8.56		
5 (15)			3.54	1.07–11.7		
6 (16–24)			4.75	1.65–13.7		
**Multivariate model**						
PICAT classes		109				0.6675
0–2			Ref		0.18	
>2–4			1.17	0.72–1.90		
≥4			1.70	0.88–3.28		
SOFA (categories)		109				
1 (0–6)			Ref		**0.0058**	
2 (7–9)			2.32	1.37–3.93		
3 (10–12)			2.45	1.26–4.75		
4 (13–14)			3.62	1.52–7.46		
5 (15)			3.43	1.03–11.5		
6 (16–24)			2.83	0.85–9.46		

Bold is use for all significant data with a *p* value < 0.05.

## Data Availability

Data are available upon request (cannot be made public because of patient confidentiality).

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
