# Peer review of "Overall Survival Rate in Allogeneic Stem Cell Transplanted Patients Requiring Intensive Care Can Be Predicted by the Prognostic Index for Critically Ill Allogeneic Transplantation Patients (PICAT) and the Sequential Organ Failure Assessment (SOFA) Scores"

_cancers, 2022, doi:10.3390/cancers14174266_

Round 1

Reviewer 1 Report

De Voeght et al. present a validation of the PICAT score, which is used to predict outcome of allogeneic HCT admitted to the ICU. This is a single-center including 111 patients admitted to the ICU between 2008 and 2018. When applying the PICAT score to their ICU population, outcome was different amongst the three predefined PICAT risk groups. I have the following comments:

·         The authors state that using the PICAT might identify patients not eligible for ICU admission because of poor outcome in light of ICU-bed scarce the COVID-19 pandemic. Based on their data, 30-day and 1-year survival of the high-risk subgroup were estimated at 31% and 13%. Please reflect on these outcome estimates in light of the COVID-19 pandemic which was the rationale of this analysis according to the introduction section. Do we need to reconsider high-risk PICAT patients for ICU admission?

·         The heterogeneity of patients admitted to the ICU post-HCT is enormous in terms of disease, transplant characteristics, but also timing after HCT to ICU admission and patients’ clinical state. Although this is similar to the original PICAT publication, the median time from HCT to ICU admission, and all components of the PICAT need to be included in Table 1.

Since most parameters are related to the transplant itself, I would suspect this model to be more suitable early after transplant than late after transplant. Please consider to validate this model in the early subset (or < median) and late subset (or > median).

·         Why did the authors not consider a c-statistic for assessment of model performance when validating PICAT?

·         After validating, the authors sought to improve outcome prediction of ICU patients with assessment of other candidate predictors. Importantly, the approach used for backward selection is rather strict (p-value <.05). Please consider a more liberal cut-off of <0.1 or even <0.2 (as used in the original PICAT publication).

·         Could the authors comment on why Platelet transfusion dependence was included? This parameter as such is not very robust because of the various causes of transfusion dependence (eg, CNS bleed for which ICU is needed, pre-engraftment, DIC, TMA, etc). What does this parameter reflect? Why not include neutrophil count as a marker of graft function?

·         Please be specific what the variable Relapse for admission denotes. Were these patients admitted with active disease (eg, relapse post-HCT) or were these patients transplanted in CR2?

·         The authors lastly propose a new model based on the multivariable analysis. Why did the authors decide for these combinations of relapse and PICAT? This needs to be explained. I strongly disagree with the final conclusion of this work that patients relapsing before ICU admission together with a high PICAT score might not benefit from ICU admission. The high risk group only consists of 3 patients, which is anecdotal. Please reconsider that part of the analysis and discussion.

·         Some predictors are captured at time of HCT while others are assessed at the time of ICU admission. Please specifically state at what time each variable was collected.

·         Please report medians instead of means when summarizing continuous variables.

·         Please state the median follow-up of patients being alive in this dataset.

·         The manuscript, tables and figures consist of a considerable amount of typographical errors and incorrect sentences. Please revise or review the manuscript with a native English speaker.

Author Response

Comments to the Author

De Voeght et al. present a validation of the PICAT score, which is used to predict outcome of allogeneic HCT admitted to the ICU. This is a single-center including 111 patients admitted to the ICU between 2008 and 2018. When applying the PICAT score to their ICU population, outcome was different amongst the three predefined PICAT risk groups. I have the following comments:

Specific comments

1) The authors state that using the PICAT might identify patients not eligible for ICU admission because of poor outcome in light of ICU-bed scarce the COVID-19 pandemic. Based on their data, 30-day and 1-year survival of the high-risk subgroup were estimated at 31% and 13%. Please reflect on these outcome estimates in light of the COVID-19 pandemic which was the rationale of this analysis according to the introduction section. Do we need to reconsider high-risk PICAT patients for ICU admission?

Re. We thank the reviewer for this comment. In order to address this important question we performed c-statistics to assess the predictive value of the PICAT (and of the SOFA) scores. We observed that although the two scores predicted survival, their predictive performances were relatively modest (c statistics for the PICAT and the SOFA scores were 0.5687 (P=0.036) and 0.6777 (P<0.0001)). Based on these findings we do not recommend to make decision foreligibility to ICU admission for allo-HCT patients based only on these two scores.We have added these data in the manuscript (see manuscript lines 351-354 and manuscript new Table 5) and discussed this issue in the discussion section of the manuscript (see manuscript lines 574-577).

2)The heterogeneity of patients admitted to the ICU post-HCT is enormous in terms of disease, transplant characteristics, but also timing after HCT to ICU admission and patients’ clinical state. Although this is similar to the original PICAT publication, the median time from HCT to ICU admission, and all components of the PICAT need to be included in Table 1.

Re. We thank the reviewer for this comment. Median time from HCT to ICU admission was 57 days. We have added this data in the manuscript (see manuscript line 249-250). Further, as suggested by the reviewerwe have added in the table 1 all components of the PICAT score (see manuscript the edited table 1).

3)Since most parameters are related to the transplant itself, I would suspect this model to be more suitable early after transplant than late after transplant. Please consider to validate this model in the early subset (or < median) and late subset (or > median).

Re. We thank the reviewer for this comment.Performing unplanned subgroup analysis is hazardous especially when subgroups are relatively small. Nevertheless, to address the reviewer comment we analyzed the impact of the PICAT score on overall survival in patients admitted to the ICU > 100 days after allo-HCT(n=43). We observe that 1-year survival was 36%, 23% and 13% in patients with low, intermediate and high PICAT score, respectively. Although the difference was not statistically significant (P=0.23) perhaps due to the low number of patients, there was no signal that the PICAT score was not predictive of survival in patients admitted > 100 days after allo-HCT. Further, we performed a proportional odds logistic regression between the three different categories of the PICAT and the period (peri-transplant vs late). We did not find a difference in the PICAT score regarding the period (p= 0.41, OR= 1.46 [0.60-3.57]).We have added these data to the manuscript (see manuscript lines 305-310.

4)Why did the authors not consider a c-statistic for assessment of model performance when validating PICAT?

Re.We thank the reviewer for this very important suggestion. According to the reviewer suggestion, we have added c-statistic calculation to assess the model performance (see manuscript lines 351-354 and manuscript new Table 5).

5)After validating, the authors sought to improve outcome prediction of ICU patients with assessment of other candidate predictors. Importantly, the approach used for backward selection is rather strict (p-value <.05). Please consider a more liberal cut-off of <0.1 or even <0.2 (as used in the original PICAT publication)?

Re.We thank the reviewer for this comment. We elected to delete this paragraph on improving the PICAT model by incorporating other factors significant in our multivariate analysis since this analysis was not very robust in regards of the relatively low patient numbers of our cohort. Instead, we have assessed whether combining the PICAT and the SOFA score would improve outcome prediction of each score. We observed that adding the PICAT to the SOFA score did not significantly improve the outcome prediction (C-statistics of 0.6784 versus 0.6777). We have added these data in the manuscript; see manuscript lines 364-365 and discussion lines 564-568. 

6)Could the authors comment on why Platelet transfusion dependence was included? This parameter as such is not very robust because of the various causes of transfusion dependence (eg, CNS bleed for which ICU is needed, pre-engraftment, DIC, TMA, etc). What does this parameter reflect? Why not include neutrophil count as a marker of graft function?

Re.We thank the reviewer for this comment. We elected to include platelet transfusion dependence in our analyses because platelet count was the strongest predictor of ICU mortality of allo-HCT patients in a recent study (PMID: 30406350, manuscript reference # 31).Univariate analyses in our cohort confirmed that low platelet count was associated with high ICU mortality. However, it was not found to be an independent factor in multivariate analysis, perhaps because of the relatively small sample size of our cohort. We elected to evaluate “independence of platelet transfusion”rather than “platelet count below or above 50 000” because it is in our opinion a more realistic observation of the severity of the thrombopenia. We have added these considerations in the discussion part of the manuscript (see manuscript lines 579-588).

7)The authors lastly propose a new model based on the multivariable analysis. Why did the authors decide for these combinations of relapse and PICAT? This needs to be explained. I strongly disagree with the final conclusion of this work that patients relapsing before ICU admission together with a high PICAT score might not benefit from ICU admission. The high risk group only consists of 3 patients, which is anecdotal. Please reconsider that part of the analysis and discussion.

Re. We thank the reviewer for this comment.We thank the reviewer for this comment and agree with it. Accordingly, we have deleted this part of the analysis from the manuscript and replace it by a comparison with the SOFA score (see also response to your comment #5).

8)Some predictors are captured at time of HCT while others are assessed at the time of ICU admission. Please specifically state at what time each variable was collected.

Re. We thank the reviewer for this comment. Parameters calculated at the time of ICU admission included biological parameters (LDH, albumin, bilirubin levels and prothrombin time-international normalized ratio) and age. We have added this information in the manuscript (see manuscript lines135-137.

9) Please report medians instead of means when summarizing continuous variables.

Re.We agree and have made these changes in the manuscript.

10)Please state the median follow-up of patients being alive in this dataset.

Re. The median follow-up is 50 months. We have added this data in the manuscript (see manuscript line 160).

11) The manuscript, tables and figures consist of a considerable amount of typographical errors and incorrect sentences. Please revise or review the manuscript with a native English speaker.

Re.We have revised the manuscript and corrected the typo and incorrect sentences.

Finally, we would like to thank the reviewer for these important constructive criticisms that helped improving substantially the manuscript.

Reviewer 2 Report

The manuscript "Early and late overall survival rate in allogeneic stem cell transplanted patients requiring intensive care can be predicted by the Prognostic Index for Critically ill Allogeneic Transplantation patients (PICAT)" is a monocentric, retrospective study involving HCT patients with ICU access between 2008 and 2018.

Major Issues:

1) Methods can be improved with the addition of a table in which the correlation between PICAT and SOFA score is explained

2) Table 1:  Add the distribution of the number and percentage of accessing the ICU for period of time (for ex: 2008-2010 = n pts/ %, 2010-2012 = etc)

3) Results: evaluate in univariate and multivariate analysis if "the  time period of ICU access"  has impact on OS 

4) Methods: add the reference to the informed consent signed by the patients for the use of the data for the analysis performed

5) Discussion: do not use term "validation". This analysis cannot be considered a validation analysis.

 Minor Issues:

1) table 1 or text: specify how many second/third HCTs  are included

2) Fig 1b: Add p value in the figure

3) Editing (see the comments in the text)

Author Response

Comments to the Author

The manuscript "Early and late overall survival rate in allogeneic stem cell transplanted patients requiring intensive care can be predicted by the Prognostic Index for Critically ill Allogeneic Transplantation patients (PICAT)" is a monocentric, retrospective study involving HCT patients with ICU access between 2008 and 2018.

Specific comments:

  1. Methods can be improved with the addition of a table in which the correlation between PICAT and SOFA score is explained.

Re. We thank the reviewer for this very important comment. Despite only two parameters are common between the SOFA and the PICAT score, we observed that the two scores were significantly correlated (see manuscript new figure 2). We further compare the ability of the 2 scores at predicting mortality using c-statistics. We observed that although the two scores predicted survival, their predictive performances were relatively modest (c statistics for the PICAT and the SOFA scores were 0.5687 (P=0.036) and 0.6777 (P<0.0001)). Further, we observed that adding the PICAT to the SOFA score did not significantly improve the outcome prediction (C-statistics of 0.6784 versus 0.6777). We have added these data in the manuscript (see manuscript lines 351-354 and manuscript new Table 5) and discussed this issue in the discussion section of the manuscript (see manuscript lines 564-577).

  1. Table 1: Add the distribution of the number and percentage of accessing the ICU for period of time (for ex: 2008-2010 = n pts/ %, 2010-2012 = etc).

Re.We thank the reviewer for this comment and have added these data in the revised table 1.

  1. Results: evaluate in univariate and multivariate analysis if "the time period of ICU access" has impact on OS.

Re. We thank the reviewer for this comment. We did not observe any significant correlation between time period of ICU admission and mortality.

Univariate Cox-regression analysis of overall survival

Variable

Categories

N

Hazard ratio

95% Confident Interval

P-value

Accessing the ICU for period of time

111

0.13

0 = 2008 à 2009

Ref

1 = 2010 à 2012

0.996

0.47-2.13

2 = 2013 à 2015

1.775

0.88-3.60

3 = 2016 à 2018

0.852

0.39-1.87

We have added these data in the manuscript (see manuscript lines 296-297).

  1. Methods: add the reference to the informed consent signed by the patients for the use of the data for the analysis performed.

Re. We thank the reviewer for this comment. This has been added in the manuscript (see manuscript lines 618-620).

  1. Discussion: do not use term "validation". This analysis cannot be considered a validation analysis.

Re. We thank the reviewer for this comment. We agree and have rephrased this sentence to “this is in concordance with data from the original publication”. See manuscript line 549.

Minor comments:
1.     table 1 or text: specify how many second/third HCTs  are included.

Re. We thank the reviewer for this comment. At the time of the first admission to ICU, 98 patients (88.3%) had received one allo-HCT, 12 patients (10.8%) had received 2 consecutive allo-HCT and 1 patient (0.9%) had received 3 consecutive allo-HCT.We have added these data in the text (see manuscript lines 241-243).

  1. Fig 1b: Add p value in the figure.

Re.We thank the reviewer for this comment and have added the p value in the legend to the figure.

  1. Editing (see the comments in the text).

Re. We thank the reviewer and have made the requested changes.

Round 2

Reviewer 1 Report

The authors have addressed my comments very well.

I have a few remaining points.

1. Since the analysis is now also focusing on SOFA score, please confirm that the SOFA score is also analyzed as a categorical variable and not continuous. The performance of a score (as measured by c-statistic) is generally better for a continuous model compared with a categorical model. Please elaborate why continuous was used if that’s true.
2. Please remove the p-value for c-statistic which is not meaningful and not correct because this p-value is from the Cox regression model. You might consider providing a confidence interval if you want to include some measure of uncertainty for the c-statistic.

3. Median follow-up time of patients alive and median time from SCT to ICU admission also require ranges.

4. The definition of relapse is still not appropriate. Please consider change to “relapse or disease progression” as patients with lymphoma or MPN are not always without signs of disease during SCT follow-up.

5. The discussion still needs a short recommendation on the clinical utility of these scores. How should we deal with these risk models in clinical practice?

Author Response

Comments to the Author

Since the analysis is now also focusing on SOFA score, please confirm that the SOFA score is also analyzed as a categorical variable and not continuous. The performance of a score (as measured by c-statistic) is generally better for a continuous model compared with a categorical model. Please elaborate why continuous was used if that’s true.

Re. We thank the reviewer for this comment. Initially, we performed the analysis with the SOFA score as a continuous variable because it leads to a better performance of the score as mentioned by reviewer. To make the comparison easier between the two scores, we performed the analysis with the use of SOFA as a categorical variable (6 different categories defined as previously reported in the original papers of the SOFA score by JL VINCENT et al. Crit Care Med.1998; see manuscript lines 102-103). As expected by the reviewer, in our univariate model, c-statistic was slightly better with continuous data (c-statistic 0.6777) than with categorical data (c-statistic 0.6577). We presented the 6 categories (see manuscript the edited table 1 and lines 145-178) and presented our new model with categorical data (see manuscript the edited table 5).

Please remove the p-value for c-statistic which is not meaningful and not correct because this p-value is from the Cox regression model. You might consider providing a confidence interval if you want to include some measure of uncertainty for the c-statistic.

Re. We thank the reviewer for this comment and we deleted the p-value from the text (see manuscript lines 238-241).

Median follow-up time of patients alive and median time from SCT to ICU admission also require ranges.

Re. We thank the reviewer for this comment. We added the median follow up for all patients and patients alive (see manuscript lines 125-127). We added the range for the median time from SCT to ICU admission (see manuscript line 178).

The definition of relapse is still not appropriate. Please consider change to “relapse or disease progression” as patients with lymphoma or MPN are not always without signs of disease during SCT follow-up.

Re. We thank the reviewer for this comment. We changed, as suggested, to “relapse of original disease or disease progression” (see manuscript lines 83-85).

The discussion still needs a short recommendation on the clinical utility of these scores. How should we deal with these risk models in clinical practice?

Re. We thank the reviewer for this comment. Our data highlighted how much clinical scores remain to be improved. Indeed, the original work of PICAT showed a better performance of PICAT compared to SOFA and we did not. But considering these data we added a short recommendation of the use of these scores in our discussion. Even if discussions between patients and their physicians remain important, we believe that patients with low PICAT and low SOFA should be admitted without any discussion and as soon as possible without any limits meanwhile (see manuscript lines 325-326).